# Pulsed Radiofrequency for Lumbosacral Radicular Pain in Dogs: Description and Assessment of an Ultrasound- and Fluoroscopy-Guided Technique in a Cadaveric Model

**DOI:** 10.3390/ani15172586

**Published:** 2025-09-03

**Authors:** Roger Medina-Serra, Francisco Gil-Cano, Francisco G. Laredo, Eliseo Belda

**Affiliations:** 1Anaesthesia and Pain Management, North Downs Specialist Referrals, Bletchingley RH1 4QP, UK; 2Programa en Ciencias Veterinarias, Escuela Internacional de Doctorado de la Universidad de Murcia, Universidad de Murcia, 30100 Murcia, Spain; 3Department of Anatomy and Comparative Pathological Anatomy, Veterinary Faculty, University of Murcia, 30100 Murcia, Spain; cano@um.es; 4Departamento de Medicina y Cirugía Animal, Facultad de Veterinaria, Universidad de Murcia, 30100 Murcia, Spain; laredo@um.es; 5Hospital Veterinario Universidad de Murcia, 30100 Murcia, Spain

**Keywords:** pulsed radiofrequency, radiofrequency treatment, lumbosacral pain, radicular pain, dorsal root ganglion, ultrasound guidance, fluoroscopy guidance, pain management, dog

## Abstract

This study investigated a technique commonly used in human medicine to manage lower back pain caused by radiculopathy (involvement of a spinal nerve). The method, called pulsed radiofrequency (PRF), aims to reduce pain by applying controlled electrical bursts to the dorsal root ganglion (DRG) of the affected nerve. In dogs, radicular pain affecting the seventh spinal nerve is considered a key indicator of lumbosacral pain, yet no procedure has been established for accurately positioning a PRF electrode at near its DRG. In this study, we used canine cadavers to test and describe a combined ultrasound and fluoroscopy approach for placing the electrode near the DRG of the seventh spinal nerve, which would enable delivering PRF treatments in living dogs. By marking the cannula tip with black Indian ink and examining tissue slices afterward, we found the electrode was approximately 2 mm from the DRG. Although the study did not evaluate safety or effectiveness in live animals, these findings suggest that ultrasound and fluoroscopy can reliably guide electrode placement in dogs. Future studies should confirm whether this method can safely and effectively relieve lumbosacral radicular pain in clinical canine patients.

## 1. Introduction

Conventional percutaneous radiofrequency (RF) ablation is an established pain management modality that induces sensory dysfunction in target nerves through thermal injury, with irreversible cellular damage occurring above 45 °C and treatment temperatures typically reaching 80 °C [1]. In contrast, pulsed radiofrequency (PRF) is a minimally invasive neuromodulation technique that delivers bursts of high-intensity electric fields while maintaining the tissue temperature below 42 °C, thereby minimising the risk of thermal injury [2,3]. The dorsal root ganglion (DRG) plays a key role in sensory transduction and nociceptive modulation, making it a primary target for neuromodulation therapies [4,5]. The mechanisms of action of PRF have been investigated through in vitro and animal models, demonstrating modulation of ion channel activity, reduction in excitatory neurotransmitter release, inhibition of pro-inflammatory cytokine release, and modulation of intracellular proteins involved in neuroplasticity and inflammation. Collectively, these effects reduce neuronal excitability and induce long-term depression of synaptic activity, ultimately contributing to pain relief [6,7,8,9].

Evidence supporting the safety and efficacy of DRG-targeted PRF for lumbar radicular pain in humans is growing. PRF can be effective as a standalone treatment or as an adjunct to transforaminal epidural steroid injections, enhancing their therapeutic effects when combined [10,11,12,13,14]. In human medicine, initial electrode placement is typically guided by fluoroscopy. Sensory stimulation is then performed to reproduce the patient’s characteristic paraesthesia or pain, allowing for positional adjustments and ensuring the DRG is within the effective range of the electromagnetic field. Motor stimulation follows to reduce the risk of intraneural electrode placement [14,15,16].

In dogs with degenerative lumbosacral stenosis, lumbosacral pain is typically managed conservatively through lifestyle modification, physical rehabilitation, pharmacological treatment, or epidural steroid injections [17]. Although a recent study identified radiculopathies as the strongest predictor of lumbosacral pain in dogs [18] a DRG-targeted PRF technique for treating canine lumbosacral radicular pain has yet to be developed. Within this treatment framework, PRF could represent a minimally invasive modality worthy of exploration before the progression to more invasive surgical approaches; however, the real-time sensory feedback approach is not feasible in veterinary patients due to the requirement for general anaesthesia, which precludes assessing patient feedback in regards experiencing paraesthesia or pain. Consequently, alternative methods are necessary to confirm electrode placement near the DRG, including anatomical landmarks, motor responses elicited by a neurostimulator, and imaging modalities such as ultrasound and fluoroscopy.

In this study, we describe an ultrasound- and fluoroscopy-guided technique to position a radiofrequency electrode tip in close proximity to the DRG of the seventh lumbar spinal nerve (L7) in canine cadavers. Our aim is to provide a replicable approach that could facilitate accuracy in PRF treatments and broaden therapeutic options for managing lumbosacral radicular pain in dogs. We hypothesise that combining ultrasound and fluoroscopy will allow for precise electrode tip placement relative to the DRG in canine cadavers.

## 2. Materials and Methods

All dogs used in this study either died or were euthanised for reasons unrelated to the study and were donated to the University of Murcia through the Donation Program of the Veterinary Faculty (PDCAVetMu). Ethical approval was granted by the Ethics Committee of Animal Experimentation (CEEA-OH) and the Committee of Biosecurity in Experimentation (CBE) of the University of Murcia (registry number ES300305440012). These cadavers were also used in other studies focused on different anatomical regions, with no overlap in the areas examined here. This approach adhered to the 3R principle (replacement, reduction, and refinement), ensuring that none of the studies interfered with or altered the others. Only dogs without history of trauma or anatomical alterations affecting the study region were used.

The study was divided into two phases. In Phase I, the objective was to become familiar with the anatomy of the lumbosacral region to develop a technique for Phase II. Phase II involved performing, describing, and assessing an ultrasound- and fluoroscopy-guided technique for delivering pulsed radiofrequency to the L7 DRG.

### 2.1. Phase I: Anatomical Study

#### 2.1.1. Ultrasonographic Study

A single canine cadaver was positioned in sternal recumbency with the pelvic limbs pulled forward, and the lumbosacral region was clipped. Using a 3–13 Hz linear transducer (SL1543, MyLab Gamma, Esaote, Florence, Italy) a comprehensive examination of the lumbosacral region was performed to identify relevant structures.

#### 2.1.2. Anatomical Dissections

After the ultrasonographic study, the cadaver was frozen in sternal recumbency at −20 °C for two days. A frozen block containing the lumbosacral region was then obtained using a band saw (Sierra cinta; Mainca, Spain) and placed at −70 °C in a freezer for an additional day. Subsequently, transverse cryosections of approximately 0.25–0.5 cm thick were obtained using a commercial high-speed bone saw/slicer equipped with an adjustable thickness guide and a laser alignment line. This technique allowed for the identification and description of relevant anatomical structures in the region of interest without altering their spatial relationships. Photographs were taken of both faces of each transverse cryosection.

### 2.2. Phase II: Technique Performance and Assessment

Six thawed adult canine cadavers were used in this phase. Due to the descriptive nature of the study, sample size calculation was not considered necessary. The number of cadavers was based on availability and aligned with similar descriptive cadaveric studies.

#### 2.2.1. Preliminary Dye Testing and Selection

To optimise needle tip marking accuracy, a preliminary assessment of different dyes was conducted. The objective was to identify a dye that produced a clear, well-defined stain while minimising tissue diffusion, which could otherwise affect measurement precision. Three dyes were tested: methylene blue (Methylthionium Chloride Proverblue 5 mg/mL; Martindale Pharma, Essex, UK), black tissue dye (Davidson Marking; Bradley Products, Bloomington, MN, USA), and a black Indian ink (30 mL Black Indian Ink; Royal Talens, Apeldoorn, The Netherlands). To simulate tissue conditions, a half-leg of lamb, obtained from a local food store, was used as a model. An insulated radiofrequency cannula (CC Straight 20-gauge, 10 cm, 10 mm tip; Boston Scientific, Valencia, CA, USA) was inserted approximately 3 cm into the tissue. Then, the stylet of the cannula was removed, immersed in the selected dye, and reintroduced into the cannula twice to stain the tissue at the distal tip. The tissue was then dissected to expose the stains, and diffusion was measured using a sliding caliper.

Methylene blue (6 × 3 mm) and black tissue dye (5 × 5 mm) showed greater diffusion, while the black Indian ink (3 × 2 mm) created the most precise and localised marking; therefore, black Indian ink was chosen for its minimal diffusion and well-defined staining.

#### 2.2.2. Ultrasound- and Fluoroscopy-Guided Technique

All procedures were performed by a single operator (RMS) with the assistance of another operator (EB), both experienced anaesthetists in locoregional techniques. Each cadaver was positioned in sternal recumbency with the pelvic limbs pulled forward, and the lumbosacral region was clipped. The same transducer used in Phase I was positioned in a sagittal plane over the caudal lumbar vertebral column and slid caudally until the spinous process of the seventh lumbar vertebra was centred in the ultrasonographic image. Subsequently, the transducer was slid laterally to visualise the “crescent-moon” appearance, formed by the bony surfaces of the L7 lamina and cranial/caudal articular processes. A slight lateral tilt and caudal rotation of the transducer facilitated a lateromedial parasagittal view, revealing the “mouse sign”, in which the “eye” represents the foraminal portion of L7; the “nose” corresponds to the L7 transverse process; and the “ear” to the lamina and caudal articular process of L7 (Figure 1).

Using an in-plane technique, an insulated radiofrequency cannula (CC Straight 20-gauge, 10 cm, 10 mm tip; Boston Scientific, Valencia, CA, USA), was advanced through the epaxial region towards the lumbosacral intervertebral foramen, positioning the cannula tip near the L7 foraminal portion (Figure 1). Placement of the cannula tip at the cranial foraminal region was assessed with fluoroscopy using both ventrodorsal and ventrolateral oblique views (Figure 2). Real-time ultrasound guidance was used to monitor cannula movements until fluoroscopic views verified the final position. Once final placement of the cannula was achieved, the stylet was removed, dipped in black Indian ink, and reintroduced twice to mark the tissue at the distal tip. The procedure was then repeated on the contralateral side. During the procedure, the operator subjectively graded the visualisation quality of bony landmarks and the cannula (“good” or “poor”). If fluoroscopic images indicated that cannula repositioning was needed, the degree of readjustment (“minor” or “major”) was recorded.

#### 2.2.3. Anatomical Dissections and Technique Assessment

Transverse cryosections were performed as described in Phase I. Anatomical sections were photographed with a ruler positioned at the same focal distance as the anatomical cut being assessed. This facilitated digital measurement of the closest distance from the black mark to the targeted dorsal root ganglia. A sliding digital caliper was used in the initial two cadavers to confirm the precision via digital methods, after which digital measurements were used for the remaining cadavers.

### 2.3. Statistical Analysis

Data were recorded in an Excel spreadsheet (Microsoft Excel for Mac, Version 16.8; Microsoft, Redmond, WA, USA) and transferred to statistical software (IBM SPSS Statistic for Windows 21.0; IBM Corp., Armonk, NY, USA). Descriptive statistics were used to summarise the data. Normality was assessed using the Shapiro–Wilk test. Results were expressed as mean and standard deviation, or median and range, as appropriate.

## 3. Results

### 3.1. Phase I: Anatomical Study

The cadaver used in this phase was a 26.5 kg male mongrel with a body condition score (BCS) 6/9 according to the World Small Animal Veterinary Association (WSAVA) Classification [19]. The ultrasonographic examination identified relevant anatomical structures, including the L7 transverse process, body, articular processes, and lamina, as well as the cranial articular process of the sacrum and the medial ileal portion. Soft tissue structures such as the thoracolumbar fascia, epaxial and hypaxial muscles, and the foraminal and extraforaminal portion of L7 were also visualised. Cryosections provided detailed anatomical information, allowing for accurate mapping of the foraminal and surrounding structures (Figure 3).

### 3.2. Phase II: Technique Performance and Assessment

#### 3.2.1. Demographics

Six canine cadavers (three mongrels, one German Shepherd dog, one Dalmatian, and one Staffordshire Terrier) were included, with a mean ± standard deviation body weight and BCS of 18.3 ± 5.3 kg and 5.7 ± 1.4, respectively. During fluoroscopy and subsequent dissections, degenerative lumbosacral changes were observed in two cadavers. These findings were further confirmed by whole-body computed tomography scans performed in the same dogs as part of unrelated studies. The degenerative changes included foraminal stenosis affecting the procedural sites in the 8th, 9th, and 11th procedures.

#### 3.2.2. Ultrasound- and Fluoroscopy-Guided Technique

The visualisation of the bony references described in Phase II as well as the visualisation of the cannula was deemed good in all 12 procedures. The “nose” (L7 transverse process) and “ear” (L7 lamina and caudal articular process) of the “mouse sign” were visualised during all procedures; however, the visualisation of the “eye” (foraminal portion of L7) was considered poor in 33% of the procedures (4/12). Cannula repositioning was required in 66% (8/12) of cases after initial ultrasound guidance, as confirmed by fluoroscopy. Most of these adjustments (6/8) were minor. In one of the cases (9th procedure) with poor visualisation of the foraminal portion of L7, lumbosacral degeneration and foraminal stenosis was present.

#### 3.2.3. Anatomical Dissections and Technique Assessment

The mean ± standard deviation distance between the dye mark and the dorsal root ganglia using the present technique was 1.96 ± 1.07 mm. No intraneural dye deposition was observed. However, in 1 out of 12 procedures, the cannula tip pierced the foraminal peridural membrane, resulting in local epidural spread. In another procedure (1/12), the dural sleeve was perforated, leading to subarachnoid dye distribution. In both instances, the dye was evident medial to the intervertebral foramen. To estimate the distance from cannula tip to the DRG in these two cases, a line was drawn along the trajectory of the needle, as indicated by the black staining through the epaxial tissues, extending to the point of bone contact at the vertebral body in the foraminal region. This contact point served as the reference for calculating the distance to the DRG.

Distances from the stain to the DRG across all procedures, and representative examples of dye distribution patterns, are illustrated in Figure 4 and Figure 5, respectively. 

## 4. Discussion

This study introduces a novel ultrasound- and fluoroscopy-guided technique for the positioning of an electrode near the L7 DRG in canine cadavers, describing a method that could be applied clinically for DRG-targeted PRF in dogs with lumbosacral radicular pain. The mean distance between the dye mark and the DRG was 1.96 ± 1.07 mm, indicating close proximity to the targeted neural structure. Importantly, no intraneural dye was observed, suggesting that the technique may allow electrode placement close enough to ensure that the DRG lies within the electromagnetic field generated by PRF, while avoiding intraneural cannula positioning and potential neural insult. Such proximity may therefore represent an appropriate balance between therapeutic effectiveness and procedural safety. However, as this investigation was conducted in a cadaveric model, further studies are needed to confirm whether the technique can reliably achieve safe electrode placement and provide effective pain relief in dogs with lumbosacral radiculopathy.

Whilst a straight sharp-tipped cannula was used in the present study, alternative cannula designs may be considered when translating this technique into clinical practice. In clinical practice, aside from technical procedural aspects, the main factors influencing cannula selection are the risks of vascular and neural iatrogenic injury.

Various cannula designs are available, including straight or curved shafts, and blunt- or sharp-tipped cannulas. Blunt cannulas were originally introduced to reduce iatrogenic vascular puncture, based on the assumption that a blunt tip would displace vessels rather than penetrate them. An early animal study supported this concept [20], particularly with large-gauge needles, and the design was further justified by the theoretical advantage of separating rather than cutting tissue planes [21]. However, these benefits have not been confirmed in human studies, where the incidence of intravascular injection has been shown to be comparable to that observed with sharp needles [21,22]. Beyond the absence of proven safety advantages, blunt-tip designs are associated with notable technical limitations. Reduced steerability and the need for greater force during advancement increase procedural difficulty and may produce more extensive tissue disruption than sharp needles [21,22,23]. As an example, vascular trauma caused by blunt-tipped needles can produce irregular rents that facilitate persistent intravascular uptake, even after repositioning [21]. Clinical studies have also reported practical complications, including bending of the cannula and failure to achieve extravascular epidural placement, with a higher overall incidence of pitfalls compared to other designs [21]. Furthermore, in the author’s experience, once resistant tissues such as the fascia are breached, the sudden loss of resistance can result in uncontrolled forward movement of the tip, raising the risk of trauma to nearby structures such as the DRG.

Both direct needle trauma and subsequent intraneural injection can lead to iatrogenic neural damage. Blunt needles have shown to reduce the likelihood of fascicular injury compared to sharp needles [20,24,25]. Although perpendicular puncture has been proposed to increase the likelihood of fascicular penetration, a study using a 90° angle did not observe perineurial disruption [25].

Experimental evidence from animal models indicates that needle puncture most commonly induces transient post-traumatic inflammation and oedema rather than structural nerve damage, which resolve within four to seven days [24,25]. The integrity of the perineurium is usually preserved, and axonal or myelin disruption is rare, only observed after intrafascicular injection and particularly associated with bupivacaine. Importantly, despite microscopic changes, functional deficits have not been observed in experimental porcine or rabbit models [24,25], although transient post-injection pain was reported in some animals [24]. Injection pressure is a relevant factor, as pressures below 20 psi are not associated with structural damage [24], as opposed to values exceeding 25 psi, which are strongly associated with nerve injury [26]. The diameter of the cannula also influences neural injury. Although the degree of local inflammation was comparable regardless of cannula size, intraneural puncture with larger cannulas commonly resulted in myelin damage and greater development of intraneural haematomas [27].

Importantly, natural and pathological variations in nerve architecture, such as differences in connective tissue, fascicle size, blood supply, and distensibility, occur within different anatomical regions of the same individual, and can occur between individuals of the same species, and across species. Such variability should be considering when translating experimental findings into clinical practice.

The variation between straight and curved shafts also warrants consideration, particularly when incorporating ultrasonography as a complementary guiding modality. Curved tips, developed to facilitate the angle of insertion of radiofrequency cannulas, may be more difficult to visualise consistently within a single imaging plane due to misalignment between the shaft and tip. For this reason, a straight cannula was selected in the present study, as it provides more reliable ultrasonographic identification. Furthermore, something very relevant for the design of our study, is that injectate delivery through a blunt cannula occurs via a small lateral opening rather than directly at the tip, which in our case, would have implied lack of reliability of the dye stain when estimating the position of the cannula’s tip. If this design were to be applied in future studies assessing injectate spread, this characteristic should be considered when planning the final positioning of the tip of the cannula prior to injection.

In human medicine, fluoroscopy is commonly used for the initial positioning of the PRF electrode near the DRG, allowing for small real-time adjustments based on sensory and motor feedback from the patient [14,15,16]. The optimal distance between the tip of the PRF electrode and the DRG remains a topic of debate. The goal is to position it close enough to maximise therapeutic benefit while avoiding excessive proximity that could cause neuronal damage due to a thermal or electric field insult [2,3,28]. An in vitro study suggests that neuronal damage may occur at distances under 0.5 mm from the electrode tip [29]. Nonetheless, the clinical application of DRG-targeted PRF has rarely been associated with significant complications, with reported adverse events being self-limiting and typically associated with transient numbness or postprocedural pain exacerbation [11].

In human procedures, sensory stimulation at 50 hertz (Hz) with a voltage threshold of ≤0.5 volts (V) and an impedance <400 Ohms (Ω) is used to evoke paraesthesia or pain in the affected dermatomes, ensuring correct positioning of the electrode’s tip. Motor threshold is generally set at 2 Hz, requiring 1.5 times the sensory threshold voltage to elicit a motor response. A negative motor response below this level is essential to minimise the risk of intraneural electrode placement [14,15,16].

In anaesthetised veterinary patients, the absence of real-time sensory feedback limits the use of human approaches that rely on this feedback for verifying electrode placement; however, motor responses to electrical stimulation may still serve as an additional tool to confirm electrode positioning. Peripheral nerve stimulation is widely used in regional anaesthesia to assist with nerve localisation, relying on stimulation intensity as a surrogate for needle-to-nerve distance. Although in clinical settings, a motor response to electrostimulation between 0.2 and 0.5 milliamperes (mA) is commonly used to indicate appropriate needle positioning during peripheral nerve blocks, these thresholds do not reliably differentiate between extraneural and intraneural placement [30,31]. A motor response at ≤0.2 mA in clinical settings is strongly associated with intraneural needle positioning, yet higher intensities (>1 mA) have also been associated with intraneural positioning in both humans and animals, underscoring the variability in stimulation thresholds. This variation is influenced by intra- and interspecies differences in intraneural and extraneural connective tissue, which affect current dispersion and nerve excitability. Additionally, regional variations in fascicle density and connective tissue composition may further modulate stimulation thresholds [30,31].

In peripheral nerve blocks, mA is typically used rather than V to standardise current intensity across varying tissue impedances [31]. By contrast, PRF procedures for lumbar radiculopathies generally rely on voltage, based on the condition that the tissue impedance remains below 400 Ω. This approach helps to prevent undesirable voltage fluctuations that could arise from impedance changes during the course of the procedure, according to Ohm’s Law (mA = V/Ω) [32].

In human DRG-targeted PRF procedures, a motor threshold of 1.5 times the sensory threshold is commonly used, with an acceptable sensory response at 0.5 V (assuming impedance remains below 400 Ω). This results in an estimated motor threshold of approximately 1.9 mA for these procedures. The trend towards a higher threshold in these procedures could be understood by considering the distinct anatomical and physiological differences between peripheral nerves and the DRG, as well as the electrode’s point of contact. In human medicine, the DRG is typically approached from the cranio-dorsal aspect of the foramen, ensuring initial contact with the sensory DRG, which lacks motor fibres [4]; therefore, when targeting the DRG, the energy delivered is focused predominantly on the sensory portion of the L7 nerve rather than the motor portion. As a result, the electrode tip may be positioned within neural tissue that does not elicit motor activity, making motor response an imperfect safeguard against intraneural positioning. Therefore, previously described motor thresholds for peripheral blocks may not serve as reliable indicators of electrode proximity to the DRG or effectively rule out intraneural placement during DRG-targeted PRF procedures in dogs.

Additionally, motor threshold extrapolation from human DRG-targeted PRF procedures should account not only for anatomical differences between species but also for variations in electrode trajectory. In contrast to the dorsal approach used in humans, our approach targets the DRG from the cranial aspect rather than the dorsal aspect. Since the DRG is positioned dorsally to the ventral motor root, this alternative trajectory could result in a more balanced engagement of both sensory and motor portions of the spinal nerve. Consequently, motor thresholds for the technique described in our study could be closer to those described for peripheral nerve blocks than those described for PRF procedures; however, this hypothesis requires further validation.

Recent advancements in regional anaesthesia highlight multimodal monitoring to enhance procedural safety. The “triple monitoring” concept [33] integrates neurostimulation, ultrasound guidance, and injection pressure monitoring to improve the accuracy of nerve localisation and mitigate the risk of intraneural injection. Ultrasound provides real-time visualisation of anatomical structures, allowing for precise needle placement and minimising the likelihood of vascular or intraneural injury. However, ultrasonographic image quality and interpretation can be affected by artefacts, patient conformation, regional anatomical or pathological variations, operator expertise, and equipment settings. These factors may result in suboptimal visualisation of the needle or nerve, particularly when targeting deep or small neural structures [33,34]. The presence of degenerative lumbosacral disease and foraminal stenosis did not appear to influence the ultrasonographic visualisation of the targeted structures. In fact, poor visualisation of the foraminal portion of the nerve was more frequently observed in cadavers without pathology. However, further studies evaluating this aspect are needed to draw definitive conclusions. Perforation of the foraminal peridural membrane occurred in two cases, one of which involved a patient with degenerative changes affecting the lumbosacral foramina. Given this single instance, it is not possible to infer a higher likelihood of peridural membrane perforation in dogs with foraminal pathology. Importantly, in this study we retained the cadavers with degenerative changes, as their inclusion provides a more realistic representation of the clinical variability encountered in practice, given that foraminal stenosis and radiculopathy are common in dogs with lumbosacral pain.

The use of ultrasound in interventional pain management is increasing, primarily as an effort to reduce reliance on ionising radiation [35]. Although fluoroscopy remains the gold standard for spinal interventions, ultrasound offers the advantage of real-time imaging without radiation exposure. Recent work on ultrasound- and fluoroscopy-guided spinal procedures in dogs found that procedural radiation exposure was on the lower end of the range reported for similar interventions in humans [36]. In our study, ultrasound played a key role in guiding cannula placement, reducing the reliance on fluoroscopy by providing real-time visualisation of relevant structures; however, fluoroscopy was still required for final adjustments and confirmation of electrode positioning. This approach aligns with the “as low as reasonably achievable” (ALARA) principle [37], as fluoroscopy use was limited to necessary confirmation rather than continuous imaging. Potential physiological or pathological variability in DRG position, particularly in patients with degenerative lumbosacral changes, may affect electrode placement and motor thresholds. Future investigations should include a broader range of canine subjects to validate the clinical application of this technique. While ultrasound can identify relevant anatomical landmarks and provide real-time guidance, image quality is influenced by multiple factors. While the visualisation of bony landmarks and the cannula was good in all cases, the foraminal portion of the seventh lumbar spinal nerve was poorly visualised in 33% of procedures, limiting the ability of ultrasound alone to confirm precise needle placement. Fluoroscopy was required in 66% of cases to adjust the cannula tip position, though most adjustments were minor. These findings indicate that while ultrasound is a valuable tool for guiding the initial trajectory, it is not a definitive imaging modality for this technique in dogs, reinforcing the necessity of fluoroscopic confirmation to ensure accurate electrode placement.

The method for verifying the final position of the needle tip via dye staining has been used previously [38]; however, while practical, presents certain limitations. The reintroduction of a dye-coated stylet into the cannula may not consistently reflect the exact needle tip position due to variability in dye distribution and tissue interaction. This could lead to slight inaccuracies when determining the exact location of the needle tip relative to the DRG. To minimise these limitations, preliminary dye testing was conducted to identify the least diffusible dye and ensure minimal spread during tissue interaction. The results confirmed that Indian ink exhibited the most localised staining, providing a more accurate representation of the needle tip position.

Moreover, while the anatomical technique used in this study allowed for measuring the distance between the dye mark and the DRG in all cases, it may have led to portions of the staining being missed if they were not captured in the analysed slices. Although the cryosections were very thin (2.5–5 mm), there remained a possibility that a section of the dye closer to the DRG was not included, potentially leading to an overestimation of the measured distance in certain cases.

While a larger sample size would have provided a more comprehensive representation of potential variations, this study aimed to introduce and describe a novel technique. As such, the findings serve as an initial reference, laying the groundwork for future studies to evaluate the technique’s feasibility and efficacy in clinical scenarios.

Future research should focus on validating the clinical effectiveness and safety of this technique in live patients and establishing optimal DRG-electrode distance, considering anatomical and degenerative variations and their impact on treatment efficacy.

Although accurate cannula placement was achieved in all procedures, both post-mortem changes and operator variability should be considered when translating this technique to clinical practice. Post-mortem changes may alter tissue architecture and impedance, potentially affecting needle insertion and ultrasonographic interpretation. Operator-related factors are also important, as variability in expertise and procedural skills may influence outcomes. Whilst standardising operators is essential for methodological robustness, exploring inter-operator differences could be a valuable focus for future studies.

## 5. Conclusions

This study presents a promising ultrasound- and fluoroscopy-guided technique for PRF placement near the L7 DRG in canine cadavers. This technique achieved placement of the electrode tip within a few millimetres of the L7 DRG, without evidence of intraneural placement, suggesting that the approach may achieve a balance between procedural safety and therapeutic effectiveness. However, as these findings are based on a cadaveric model, they should be considered preliminary, and further studies in live dogs are required to confirm the safety and effectiveness of this approach, ensuring DRG-targeted PRF becomes a viable option for managing lumbosacral radicular pain in veterinary medicine.

## Figures and Tables

**Figure 1 animals-15-02586-f001:**
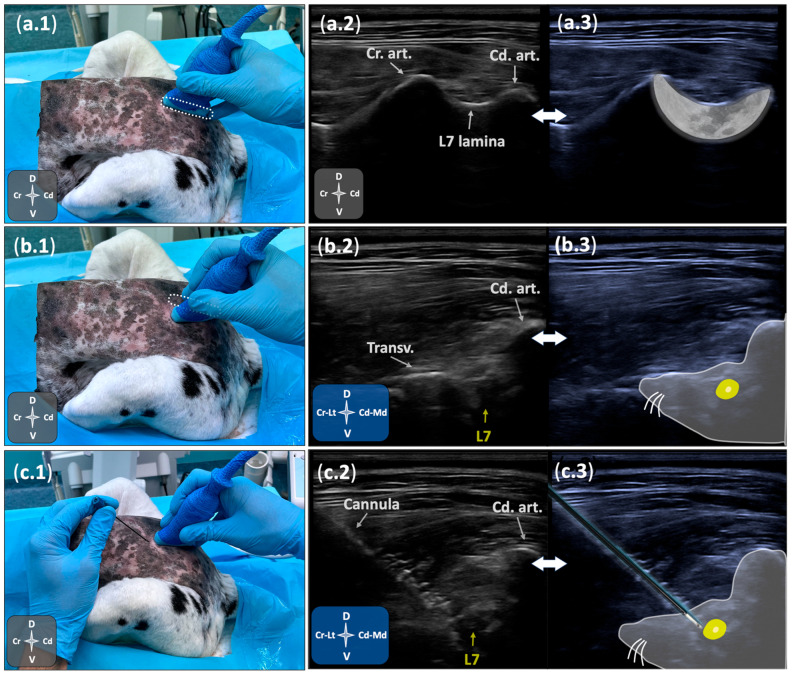
Ultrasound-guided technique for positioning a radiofrequency cannula near the L7 DRG in a canine cadaver. (**a.1**–**c.1**) Transducer positioning over the lumbosacral region in sternal recumbency with pelvic limbs extended cranially. (**a.2,a.3**) Parasagittal ultrasonographic view showing the L7 lamina and articular processes, generating the “crescent-moon” appearance. (**b.2,b.3**) Lateromedial parasagittal ultrasonographic view depicting the transverse process (“nose”), caudal articular process (“ear”), and the foraminal portion of the L7 spinal nerve (“eye”), comprising the “mouse sign”. (**c.2**,**c.3**) Ultrasound image during in-plane cannula advancement towards the foraminal portion of L7 at the cranial aspect of the intervertebral foramen. Cd. art., caudal articular process; Cr. art., cranial articular process; L7, seventh spinal nerve; Transv., transverse process. See orientation landmarks as follows: Cd, caudal; Cr, cranial; D, dorsal; Lt, lateral; Md, medial; V, ventral.

**Figure 2 animals-15-02586-f002:**
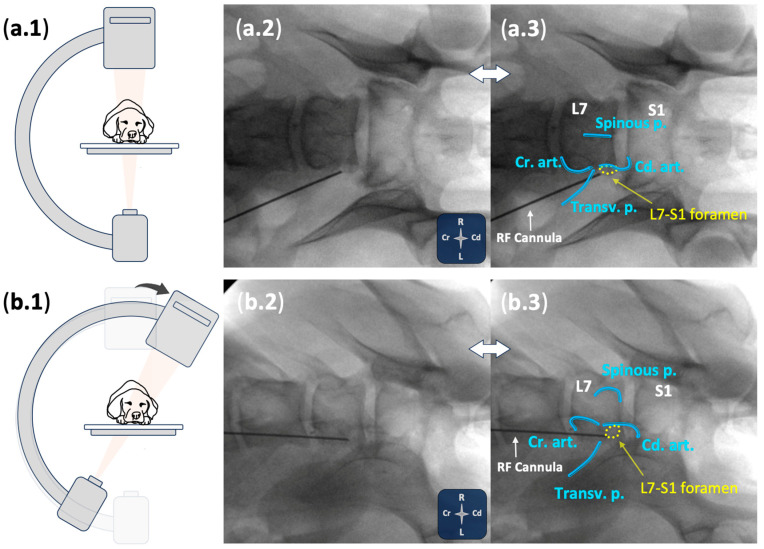
Fluoroscopic-guided assessment of cannula positioning at the cranial aspect of the L7-S1 foramen in a canine cadaver. (**a.1,b.1**) Schematic representation of C-arm positioning for ventrodorsal (**a.1**) and ventrolateral oblique (**b.1**) imaging. (**a.2**,**b.2**) Corresponding fluoroscopic images showing the trajectory and placement of the radiofrequency (RF) cannula. (**a.3,b.3**) Annotated views highlighting anatomical landmarks: Cd. art., caudal articular process; Cr. art., cranial articular process; L7, seventh lumbar vertebra; L7–S1 foramen (yellow dashed oval); S1, first sacral vertebra; Spinous p., spinous process; Transv. p., transverse process. See orientation landmarks as follows: Cd, caudal; Cr, cranial; L, left; R, right.

**Figure 3 animals-15-02586-f003:**
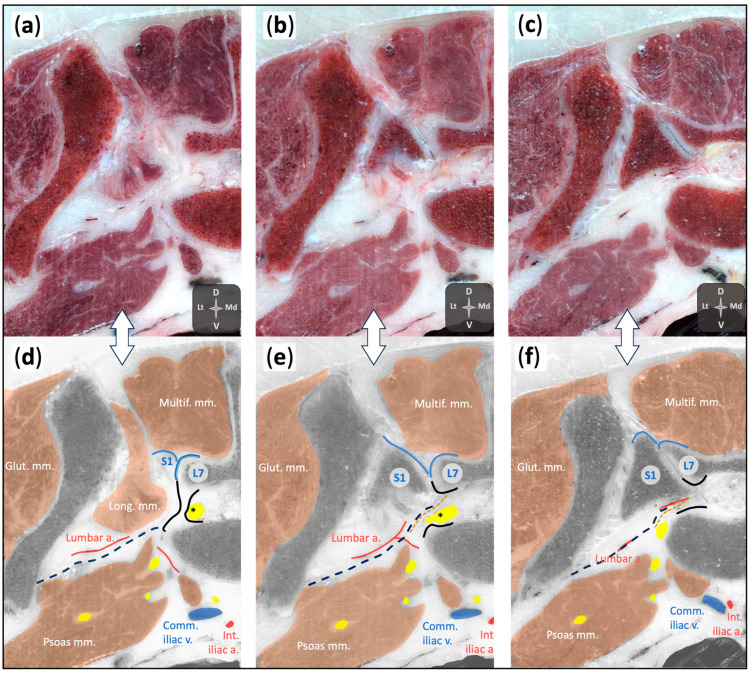
Cryosections of the lumbosacral region in a canine cadaver (**a**–**c**), progressing from cranial (**a**) to caudal (**c**), and their corresponding schematic representations (**d**–**f**). Panels (**d**), (**e**), and (**f**) correspond to panels (**a**), (**b**), and (**c**), respectively. In the schematic views, neural tissue is highlighted in yellow, and the dorsal root ganglia (DRG) of the seventh spinal nerve is indicated by a black asterisk. Comm. Iliac v., common iliac vein; Glut. mm., gluteal muscles, Int. iliac a., internal iliac artery; L7, seventh lumbar vertebra; Lumbar a., lumbar artery; Multif. mm., multifidus muscle; Psoas mm., psoas muscle S1, first sacral vertebra. See orientation landmarks as follows: D, dorsal; Lt, lateral; Md, medial; V, ventral.

**Figure 4 animals-15-02586-f004:**
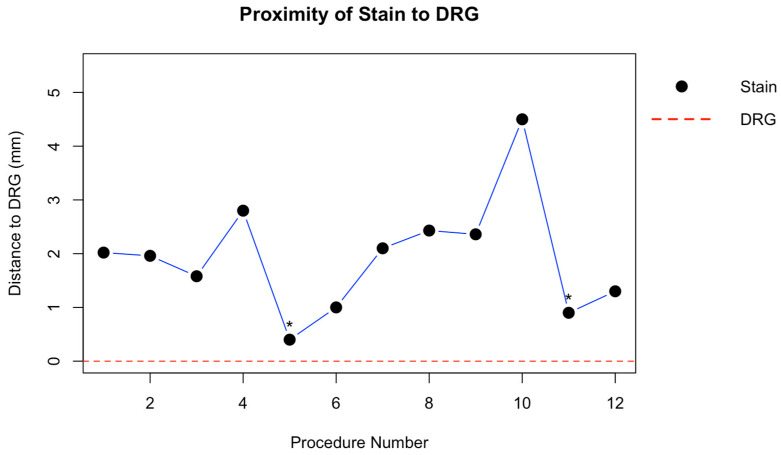
Distance from the dye mark to the dorsal root ganglion (DRG) following ultrasound and fluoroscopy guided cannula placement in canine cadavers. Each black dot represents an individual procedure (*n* = 12), with measurements obtained from transverse cryosections. In the two cases where dye extended medially, due to epidural spread (11th procedure) or subarachnoid spread (5th procedure), the distance was estimated by extrapolating the needle trajectory to its point of bony contact within the foraminal region. Asterisks denote the two cases with epidural and subarachnoid dye uptake.

**Figure 5 animals-15-02586-f005:**
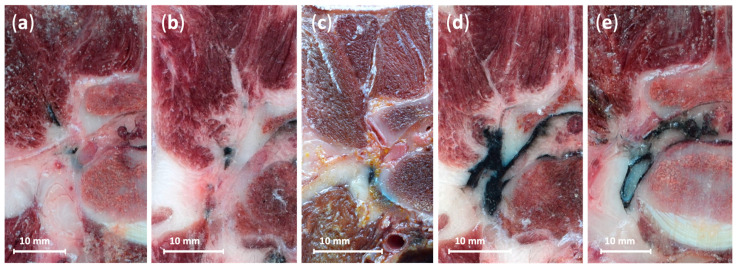
Representative cryosections showing dye distribution patterns following radiofrequency cannula placement in canine cadavers. Panels (**a**) (6th procedure), (**b**) (8th procedure), and (**c**) (1st procedure, mirrored for consistency) illustrate the three most common staining patterns, with black Indian ink marking the target region adjacent to the L7 dorsal root ganglion. Panels (**d**) (11th procedure) and (**e**) (5th procedure, mirrored for consistency) show the two atypical outcomes observed across the 12 procedures: (**d**) depicts local epidural spread due to perforation of the foraminal peridural membrane, and (**e**) shows subarachnoid spread resulting from inadvertent penetration of the dural sleeve. Scale bars = 10 mm. See orientation landmarks as follows: D, dorsal; Lt, lateral; Md, medial; V, ventral.

## Data Availability

Data supporting the reported results can be sent to anyone interested by contacting the corresponding author.

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
