# Peer review of "Pulsed Radiofrequency for Lumbosacral Radicular Pain in Dogs: Description and Assessment of an Ultrasound- and Fluoroscopy-Guided Technique in a Cadaveric Model"

_animals, 2025, doi:10.3390/ani15172586_

Round 1
Reviewer 1 Report
Comments and Suggestions for Authors
Dear Authors,
Thank you for submitting this interesting and relevant manuscript describing a novel ultrasound- and fluoroscopy-guided technique for pulsed radiofrequency electrode placement targeting the L7 dorsal root ganglion in canine cadavers. The study is original and it represents significant step forward in veterinary interventional pain management. The manuscript is well-structured, with only few minor revisions and typographical and formatting issues, which can be easily corrected during revision and editing. Below, my detailed comments:
Line 24: I would suggest replacing “about” with “approximately”.
Line 67: missing space and punctuation. “...dogs [16]. A DRG-targeted PRF…”
Line 84-86: These statements seem more appropriate for the limitations section of the Discussion or reiterated in the Conclusions.
Line 97-98: I would suggest adding “in the lumbosacral region”.
Line 143: Were all procedures performed by the same single operator? If so, do you expect significant variability if multiple clinicians attempt the same approach?
Line 154: Considering that the cannula used had a straight, non-blunt tip and that repositioning was required in 66% (8/12) of procedures after initial ultrasound guidance, is there a potential risk of iatrogenic injury to vascular or neural structures during cannula placement? Could you comment on why this particular cannula design was chosen over commercially available curved or blunt-tipped cannulas?
Line 157-162: Did you record the time needed to complete the procedure? It may have been interesting to quantify radiation exposure, especially for risk–benefit analysis in future clinical applications.
Line 209: the reference is missing.
Line 231-233: It seems to me there is a discrepancy between the Methods (line 97: “only dogs without history of trauma or anatomical alterations were used”) and the Results, where two cadavers are reported to have degenerative lumbosacral disease with foraminal stenosis. I understand that such pathology may not have been detectable before, however, this should be clarified for the reader. Specifically, why the CT scan was performed? And when? Why were these two cadavers not excluded from the study after detection of degenerative changes?
Even if their inclusion was intentional (for example, to reflect clinical variability), it might be important to explicitly explain it to the reader.
Line 267: “Figure 5 .” extra-space.
Line 347-349: When discussing approaches to avoid intraneural needle placement, it might be worth also mentioning the potential role of needle bevel type, as long bevel-needle tips (such as the tip of the cannula used in this study) are more likely to pierce the perineurium than blunt tips.
Line 360: Do you think that using a different cannula tip would have led to different results?
Out of scientific curiosity, could you please explain which are the clinical implications of perforating the peridural membrane during pulsed radiofrequency procedures? This could be an interesting point to briefly discuss if the authors consider it relevant for future clinical application.
Line 388: Typo (“or” instead of “for”) and missing space before [27].
Line 388-406: While the authors acknowledge some limitations, the manuscript would benefit from addressing two (maybe obvious) additional aspects: the operator-related variability and the cadaver versus live patient differences. All procedures seem to have been performed by a single operator, therefore results may vary among clinicians with different levels of expertise, which could influence reproducibility. In addition, tissue properties in cadavers may not fully reflect live patient conditions, potentially affecting needle trajectory, imaging visibility, and technique accuracy.
References: The font is not consistent with the rest of the manuscript. There are few DOI formatting inconsistencies (some haver periods after DOI, others don’t).
Author Response
Dear reviewer, many thanks for your time and effort during the revision process. Your review has been key for improving clarity and value of the manuscript. Please find answered your comments one by one below. We hope we have addressed your comments and concerns.
We sincerely thank you for your time and effort throughout the revision process. Your thoughtful feedback has been very valuable in improving the clarity and overall quality of the manuscript. Please, find below the answers to your comments, one by one. We hope to have addressed the key aspects you highlighted.
Dear Authors,
Thank you for submitting this interesting and relevant manuscript describing a novel ultrasound- and fluoroscopy-guided technique for pulsed radiofrequency electrode placement targeting the L7 dorsal root ganglion in canine cadavers. The study is original and it represents significant step forward in veterinary interventional pain management. The manuscript is well-structured, with only few minor revisions and typographical and formatting issues, which can be easily corrected during revision and editing. Below, my detailed comments:
Line 24: I would suggest replacing “about” with “approximately”.
Amended.
Line 67: missing space and punctuation. “...dogs [16]. A DRG-targeted PRF…”
Thanks for spotting this. Amended.
Line 84-86: These statements seem more appropriate for the limitations section of the Discussion or reiterated in the Conclusions.
Thank you so much for this comment. We have now removed that section from the introduction and incorporated relevant content in the discussion (line 287-296) and reiterated it in the conclusion (line 496-502).
Line 97-98: I would suggest adding “in the lumbosacral region”.
We have now specified “No history of trauma or anatomical alterations affecting the study region...”
Line 143: Were all procedures performed by the same single operator? If so, do you expect significant variability if multiple clinicians attempt the same approach?
Thanks for this comment. As the reviewer suggests, a degree of variation amongst operators when performing the technique could potentially lead to differing outcomes. To control the potential operator variability, in our study, all procedures were performed by the same operator. This has now been specified in line 149-150 “All injections were performed by a single operator (RMS) with the assistance of another operator (EB), both experienced anaesthetist in locoregional techniques.”
The discussion has been amended to acknowledge that variability in expertise and procedural skills may affect outcomes. While standardising operators should be a reasonable option in future studies to strengthen their methodology, inter-operator differences could be an informative focus for future studies. Amendments in line 490-493.
Line 154: Considering that the cannula used had a straight, non-blunt tip and that repositioning was required in 66% (8/12) of procedures after initial ultrasound guidance, is there a potential risk of iatrogenic injury to vascular or neural structures during cannula placement? Could you comment on why this particular cannula design was chosen over commercially available curved or blunt-tipped cannulas?
We sincerely thank the reviewer for this insightful comment, which has prompted us to expand our discussion (line 298-355), and we believe has enhanced both the clarity and quality of the manuscript.
Line 157-162: Did you record the time needed to complete the procedure? It may have been interesting to quantify radiation exposure, especially for risk–benefit analysis in future clinical applications.
We did not record the time required to complete the procedure in this study. However, we have recently published an article on radiation exposure during spinal interventional pain management procedures, in which the technique described in the cadaveric model has been widely applied. We believe this work may be relevant to the reviewer’s consideration of risk-benefit aspects in regards its clinical application.
https://www.vaajournal.org/article/S1467-2987(25)00147-3/abstract
Line 209: the reference is missing.
Thanks for spotting this. Amended.
Line 231-233: It seems to me there is a discrepancy between the Methods (line 97: “only dogs without history of trauma or anatomical alterations were used”) and the Results, where two cadavers are reported to have degenerative lumbosacral disease with foraminal stenosis. I understand that such pathology may not have been detectable before, however, this should be clarified for the reader. Specifically, why the CT scan was performed? And when? Why were these two cadavers not excluded from the study after detection of degenerative changes?
Even if their inclusion was intentional (for example, to reflect clinical variability), it might be important to explicitly explain it to the reader.
We thank the reviewer for highlighting to us this lack of clarity regarding the use of CT in this study. During fluoroscopy and subsequent dissections, degenerative changes suggestive of lumbosacral disease were observed in two cadavers. We took advantage of whole-body CT scans, which had been performed in these dogs as part of other unrelated studies, to confirm the presence of foraminal stenosis. These findings were therefore incidental and not known before the procedures. We chose to retain these cadavers in the analysis, as their inclusion provides a more realistic representation of the clinical variability encountered in practice.
These considerations have been clarified in the revise manuscript in line 236-240 and 486-489.
Line 267: “Figure 5 .” extra-space.
The extra space between “Figure” and “5” appears to result from text justification in the manuscript file. We trust that the editorial team will be able to correct this during the typesetting process.
Line 347-349: When discussing approaches to avoid intraneural needle placement, it might be worth also mentioning the potential role of needle bevel type, as long bevel-needle tips (such as the tip of the cannula used in this study) are more likely to pierce the perineurium than blunt tips.
In line with the previous comment, we have now expanded the discussion in regards cannula type and its implications (line 298-355).
Line 360: Do you think that using a different cannula tip would have led to different results?
Out of scientific curiosity, could you please explain which are the clinical implications of perforating the peridural membrane during pulsed radiofrequency procedures? This could be an interesting point to briefly discuss if the authors consider it relevant for future clinical application.
We thank the reviewer for this interesting comment. In our view, perforation of the peridural membrane does not require additional consideration, as the critical factor is the distance between the electrode tip and the neural tissue rather than its relationship to other fascial layers. As discussed in the manuscript, proximity to the neural target when applying this technique in the clinical setting is guided not only by ultrasound and fluoroscopy but also by neurostimulation, which ultimately helps guiding the final distance from the electrode tip to the neural structures before delivering PRF.
Line 388: Typo (“or” instead of “for”)
Thanks for spotting this. Amended.
and missing space before [27].
In line with the comment above, this error appears to result from text justification in the manuscript.
Line 388-406: While the authors acknowledge some limitations, the manuscript would benefit from addressing two (maybe obvious) additional aspects: the operator-related variability and the cadaver versus live patient differences. All procedures seem to have been performed by a single operator, therefore results may vary among clinicians with different levels of expertise, which could influence reproducibility. In addition, tissue properties in cadavers may not fully reflect live patient conditions, potentially affecting needle trajectory, imaging visibility, and technique accuracy. In addition, tissue properties in cadavers may not fully reflect live patient conditions, potentially affecting needle trajectory, imaging visibility, and technique accuracy.
Thanks for this comment. In regards inter-operator variability, please refer to the related answer in a previous comment. In regards cadaveric nature of the study, we have now acknowledged that cadaver vs life patient model may influence technique outcomes and this should be considered when translating this technique to clinical practice (line 468-475).
References: The font is not consistent with the rest of the manuscript. There are few DOI formatting inconsistencies (some haver periods after DOI, others don’t).
We thank the reviewer for this observation. The references section has been reformatted, and the font is now consistent throughout the manuscript. With respect to the DOI entries, we carefully reviewed the reference list and did not identify any instances where the DOI was presented without a period at the end.
Reviewer 2 Report
Comments and Suggestions for Authors
This study marks a significant advance in the management of radicular pain in dogs, but the findings should be considered preliminary. While the technique shows promise, further research is needed to confirm its clinical utility and establish protocols to ensure its safety and efficacy when used on live patients. As a preliminary anatomical study, it is sound, but the title should make it clear that the study is preliminary. It should be noted that only six cadavers were used in the study, which may not be representative of existing anatomical variability.
Note: Line 235 should read 3.2.2; and line 246 should read 3.2.3.
Author Response
Thank you for the time and effort spent during the revision of the present manuscript. Your comments have helped us strengthen the clarity for the readers.
This study marks a significant advance in the management of radicular pain in dogs, but the findings should be considered preliminary. While the technique shows promise, further research is needed to confirm its clinical utility and establish protocols to ensure its safety and efficacy when used on live patients. As a preliminary anatomical study, it is sound, but the title should make it clear that the study is preliminary. It should be noted that only six cadavers were used in the study, which may not be representative of existing anatomical variability.
Many thanks for this insightful comment. It has now been emphasised in the conclusion that the results should be considered as preliminary in line 500.
Note: Line 235 should read 3.2.2; and line 246 should read 3.2.3.
Many thanks for spotting this. Amended.
Reviewer 3 Report
Comments and Suggestions for Authors
I would like to thank the authors for the opportunity to review this well-structured and interesting manuscript. The topic is highly relevant and addresses a notable gap in the current veterinary literature regarding image-guided interventions for managing lumbosacral radicular pain in dogs. The study is clearly written, the methodology is robust, and the figures enhance the understanding of the proposed technique. I appreciate the authors’ effort to explore innovative solutions in veterinary pain management and offer the following comments and suggestions for minor revisions aimed at strengthening the final version of the manuscript.
The current title is appropriate and reflects the content of the manuscript clearly. It accurately conveys the species involved, the anatomical target , and the method (pulsed radiofrequency under ultrasound and fluoroscopic guidance).
Introduction: While the introduction sets the context well and explains the rationale for targeting the L7 DRG, it would benefit from more clearly highlighting the potential clinical role of this technique in managing chronic pain (line 67-69) in particular of L7 in dogs. It would also be helpful to acknowledge that, although other treatment options exist—such as pharmacological approaches (e.g., a recently published article on the use of oral amantadine in dogs for this type of pain- 10.1186/s12917-025-04911-9)—no gold standard treatment currently exists for canine radiculopathy. Therefore, ongoing research into alternative and interventional strategies like PRF remains important. Please consider citing the recent study on amantadine in dogs in the bibliography to support this point.
In addition, I recommend including the following reference: Ball RD (2014). The science of conventional and water-cooled monopolar lumbar radiofrequency rhizotomy: an electrical engineering point of view. Pain Physician, 17, E175–E211. This publication, along with work by Cosman (already cited), provides an important biophysical explanation of PRF mechanisms and should be cited in support of the conceptual background.
Materials and Methods: This section is detailed and well-structured, allowing for clear understanding and reproducibility of the procedure. The use of both ultrasound and fluoroscopic guidance is appropriate and well justified. The accompanying images are clear and useful for visualizing anatomical landmarks and procedural steps. One suggestion for improvement: please provide more detail on how the slicing of cryosections was performed. Was a physical guide or slicing jig used to ensure consistent sectioning? This clarification would enhance methodological transparency.
At Line 209 (reference): Typographical error – please revise.
Results: The results are clearly presented and support the main objective of demonstrating technical feasibility.
Discussion: The discussion remains appropriately cautious, reflecting the cadaveric nature of the study while still highlighting its translational potential. The interpretation is well aligned with the results.
I agree with the authors’ hypothesis that future clinical applications may be more challenging due to anatomical alterations associated with lumbosacral stenosis. In such cases, it is reasonable to assume that ultrasound may face limitations in image resolution or landmark identification, potentially making fluoroscopy a more reliable imaging modality for guidance in pathological patients.
Additionally, as a practical suggestion: at Lines 369–387:
Have you considered that one millilitre of non-ionic iodinated radiographic contrast agents contrast could be administered in the final view to assess the absence of epidural spread and to exclude any intravascular placement?
This could further enhance confidence in the correct placement of the electrode.
In conclusion, this manuscript presents a promising and technically sound method for PRF electrode placement in dogs, with potential applications in treating chronic lumbosacral pain. The study is methodologically robust, the writing is clear, and the imaging support is strong. With minor clarifications and the addition of a few relevant references, the article will make a valuable contribution to veterinary neuromodulation research.
Author Response
Many thanks for the time and effort spent during the review process and for your valuable comments.
Your comments will be addressed individually below:
I would like to thank the authors for the opportunity to review this well-structured and interesting manuscript. The topic is highly relevant and addresses a notable gap in the current veterinary literature regarding image-guided interventions for managing lumbosacral radicular pain in dogs. The study is clearly written, the methodology is robust, and the figures enhance the understanding of the proposed technique. I appreciate the authors’ effort to explore innovative solutions in veterinary pain management and offer the following comments and suggestions for minor revisions aimed at strengthening the final version of the manuscript.
The current title is appropriate and reflects the content of the manuscript clearly. It accurately conveys the species involved, the anatomical target, and the method (pulsed radiofrequency under ultrasound and fluoroscopic guidance).
Introduction: While the introduction sets the context well and explains the rationale for targeting the L7 DRG, it would benefit from more clearly highlighting the potential clinical role of this technique in managing chronic pain (line 67-69) in particular of L7 in dogs. It would also be helpful to acknowledge that, although other treatment options exist—such as pharmacological approaches (e.g., a recently published article on the use of oral amantadine in dogs for this type of pain- 10.1186/s12917-025-04911-9)—no gold standard treatment currently exists for canine radiculopathy. Therefore, ongoing research into alternative and interventional strategies like PRF remains important. Please consider citing the recent study on amantadine in dogs in the bibliography to support this point.
Thank you for this comment. We have revised the Introduction to provide clearer context on the current management of canine lumbosacral pain in line 71-76: “In dogs with degenerative lumbosacral stenosis, lumbosacral pain is typically managed conservatively through lifestyle modification, physical rehabilitation, pharmacological treatment, or epidural steroid injections (17). Surgical decompression is generally reserved for cases unresponsive to conservative measures or when significant neurological deficits are present”. “Within this treatment framework, PRF could represent a minimally invasive modality worthy of exploration before progression to more invasive surgical approaches.”
We also appreciate the reference to the recent study on amantadine in dogs. While this provides useful insight into pharmacological management, we will cite the review by Worth et al. (2019), which provides a comprehensive overview of current management approaches for lumbosacral pain in dogs and directly supports the rationale for exploring interventional options such as PRF.
In addition, I recommend including the following reference: Ball RD (2014). The science of conventional and water-cooled monopolar lumbar radiofrequency rhizotomy: an electrical engineering point of view. Pain Physician, 17, E175–E211. This publication, along with work by Cosman (already cited), provides an important biophysical explanation of PRF mechanisms and should be cited in support of the conceptual background.
Thanks for this comment. We have now introduced conventional radiofrequency and cited the aforementioned reference in line 48-51.
Materials and Methods: This section is detailed and well-structured, allowing for clear understanding and reproducibility of the procedure. The use of both ultrasound and fluoroscopic guidance is appropriate and well justified. The accompanying images are clear and useful for visualizing anatomical landmarks and procedural steps. One suggestion for improvement: please provide more detail on how the slicing of cryosections was performed. Was a physical guide or slicing jig used to ensure consistent sectioning? This clarification would enhance methodological transparency.
Thanks for this thorough comment. We have now specified (line 120) that the saw was equipped with an adjustable thickness guide and a laser alignment.
At Line 209 (reference): Typographical error – please revise.
Thanks for spotting this. Amended.
Results: The results are clearly presented and support the main objective of demonstrating technical feasibility.
Thank you.
Discussion: The discussion remains appropriately cautious, reflecting the cadaveric nature of the study while still highlighting its translational potential. The interpretation is well aligned with the results. I agree with the authors’ hypothesis that future clinical applications may be more challenging due to anatomical alterations associated with lumbosacral stenosis. In such cases, it is reasonable to assume that ultrasound may face limitations in image resolution or landmark identification, potentially making fluoroscopy a more reliable imaging modality for guidance in pathological patients.
Additionally, as a practical suggestion: at Lines 369–387:
Have you considered that one millilitre of non-ionic iodinated radiographic contrast agents contrast could be administered in the final view to assess the absence of epidural spread and to exclude any intravascular placement?
This could further enhance confidence in the correct placement of the electrode.
Thank you so much for this consideration. Indeed, your consideration is the rationale for a present complementary study, in which after cannula positioning, we inject a contrast-dye solution to assess the spread of the injectate.
In conclusion, this manuscript presents a promising and technically sound method for PRF electrode placement in dogs, with potential applications in treating chronic lumbosacral pain. The study is methodologically robust, the writing is clear, and the imaging support is strong. With minor clarifications and the addition of a few relevant references, the article will make a valuable contribution to veterinary neuromodulation research.
Many thanks for your constructive feedback and your considerations.